# RandAugment: Practical Automated Data Augmentation with a Reduced Search Space

**Ekin D. Cubuk,**\* **Barret Zoph,**\* **Jonathon Shlens,  Quoc V. Le**
Google Research, Brain Team
{cubuk,barretzoph,shlens,qvl}@google.com

## Abstract

Recent work on automated data augmentation strategies has led to state-of-the-art results in image classification and object detection. An obstacle to a large-scale adoption of these methods is that they require a separate and expensive search phase. A common way to overcome the expense of the search phase was to use a smaller proxy task. However, it was not clear if the optimized hyperparameters found on the proxy task are also optimal for the actual task. In this work, we rethink the process of designing automated data augmentation strategies. We find that while previous work required searching for many augmentation parameters (e.g. magnitude and probability) independently for each augmentation operation, it is sufficient to only search for a single parameter that jointly controls all operations. Hence, we propose a search space that is vastly smaller (e.g. from $10^{32}$ to $10^2$ potential candidates). The smaller search space significantly reduces the computational expense of automated data augmentation and permits the removal of a separate proxy task. Despite the simplifications, our method achieves state-of-the-art performance on CIFAR-10, SVHN, and ImageNet. On EfficientNet-B7, we achieve 84.7% accuracy, a 1.0% increase over baseline augmentation and a 0.4% improvement over AutoAugment on the ImageNet dataset. On object detection, the same method used for classification leads to 1.0-1.3% improvement over the baseline augmentation method on COCO. Code is available online. [2] [3]

## 1   Introduction

Although data augmentation is a widely used method to add additional knowledge when training vision models [30, 15, 5, 42], the fact that it is manually designed makes it difficult to scale to new applications. Learning data augmentation strategies from data has recently emerged as a new paradigm to automate the design of augmentation and has the potential to address some weaknesses of traditional data augmentation methods [3, 45, 13, 17]. Training a machine learning model with a learned data augmentation policy may significantly improve image classification [3, 17, 13], object detection [45], model robustness [23, 39, 28], and semi-supervised learning for image classification [37]. Unlike architecture search [47], all of these improvements in predictive performance incur no additional computational cost at inference time since data augmentation is only used during training.

An obstacle to a large-scale adoption of these methods is that they require a separate and expensive search phase. A common way to overcome the expense of the search phase was to use a smaller proxy task. Although the proxy task helps speeding up the search process, it also adds extra complexity to the methods and causes further issues. For example, it was not clear if the optimal hyperparameters found on the proxy task are also optimal for the actual task. In fact, we will provide experimental

| | search space | CIFAR-10 PyramidNet | SVHN Wide-ResNet | ImageNet EfficientNet-B7 |
|---|---|---|---|---|
| Baseline | 0 | 97.3 | 98.5 | 83.7 |
| AA | $10^{32}$ | 98.5 | 98.9 | 84.3 |
| Fast AA | $10^{32}$ | 98.3 | 98.8 | - |
| PBA | $10^{61}$ | 98.5 | 98.9 | - |
| Adv. AA | $10^{32}$ | **98.6** | - | - |
| RA (ours) | $10^{2}$ | 98.5 | **99.0** | **84.7** |

Table 1: **Simple grid search on a vastly reduced search space matches or exceeds predictive performance of other augmentation methods.** We report the search space size, and the test accuracy achieved for AutoAugment (AA) [3], Fast AutoAugment [17], Population Based Augmentation (PBA) [13], Adversarial AutoAugment [43] and the proposed RandAugment (RA) on CIFAR-10 [14], SVHN [24], and ImageNet [4] classification tasks. Search space size is reported as the order of magnitude of the number of possible augmentation policies. Dash indicates that results are not available.

evidence in this paper to challenge this core assumption. In particular, we demonstrate that using proxy tasks is sub-optimal as the strength of the augmentation depends strongly on model and dataset size. These results suggest that improved data augmentation methods may be possible if one could remove the separate search phase on a proxy task.

In this work, we aim to make AutoAugment and related methods [3, 13, 17] more practical. While previous work focused on the search methodology [17, 13], our analysis shows that the search space plays a more significant role. In previous work, it was required to search for both the probability and the magnitude (e.g. how many degrees to rotate an image) of each operation in the search space. Our experiments surprisingly show that it is sufficient to optimize all of the operations jointly with a single distortion magnitude. To further simplify the search space we simply set the probability of each operation to be equally likely. With the reduced search space, we also simplify the whole search process: we no longer need a separate expensive search phase and proxy tasks.

The reduction in parameter space is in fact so dramatic that a simple grid search is sufficient to find a data augmentation policy that outperforms or closely matches AutoAugment. We name our method RandAugment because for each image it uniformly samples from all available data augmentation operations in the search space. Table 1 shows a summary of our main results, which reveals RandAugment can be easily optimized thanks to having a much smaller search space. RandAugment also achieves higher accuracy on a wide range of benchmarks, thanks to its ability to adjust its distortion magnitude accordingly to the the model and dataset size. With EfficientNet-B7, we achieve an accuracy of 84.7%, a 0.4% increment over AutoAugment and 1.0% over baseline augmentation of flips and crops [32].

## 2   Related Work

Data augmentation has played a central role in the training of deep vision models. On natural images, horizontal flips and random cropping or translations of the images are commonly used in classification and detection models [41, 15, 9]. On MNIST, elastic distortions across scale, position, and orientation have been applied to achieve impressive results [30, 2, 36, 29]. While previous examples augment the data while keeping it in the training set distribution, operations that do the opposite can also be effective in increasing generalization. Some methods randomly erase or add noise to patches of images for increased validation accuracy [6, 44], robustness [33, 39, 7], or both [23]. Mixup [42] is a particularly effective augmentation method on CIFAR-10 and ImageNet, where the neural network is trained on convex combinations of images and their corresponding labels.

Moving away from individual operations to augment data, other work has focused on finding optimal strategies for combining different operations. For example, Smart Augmentation learns a network that merges two or more samples from the same class to generate new data [16]. Tran et al. generate augmented data via a Bayesian approach, based on the distribution learned from the training set [35]. DeVries et al. use transformations (e.g. noise, interpolations and extrapolations) in the learned feature space to augment data [5]. Furthermore, generative adversarial networks (GAN) have been used to choose optimal sequences of data augmentation operations[27].

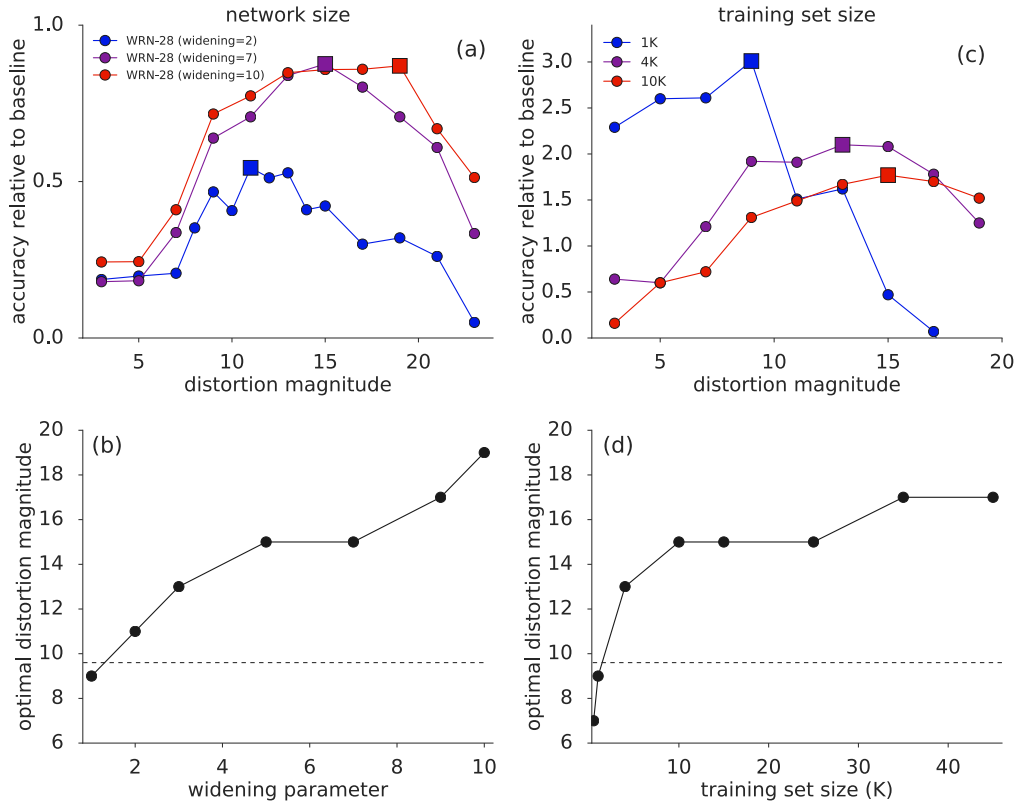

Figure 1: **Optimal magnitude of augmentation depends on the size of the model and the training set.** All results report CIFAR-10 validation accuracy for Wide-ResNet model architectures [41] averaged over 20 random initializations, where $N = 1$. (a) Accuracy of Wide-ResNet-28-2, Wide-ResNet-28-7, and Wide-ResNet-28-10 across varying distortion magnitudes. Models are trained for 200 epochs on 45K training set examples. Squares indicate the distortion magnitude that achieves the maximal accuracy. (b) Optimal distortion magnitude across 7 Wide-ResNet-28 architectures with varying widening parameters ($k$). (c) Accuracy of Wide-ResNet-28-10 for three training set sizes (1K, 4K, and 10K) across varying distortion magnitudes. Squares indicate the distortion magnitude that achieves the maximal accuracy. (d) Optimal distortion magnitude across 8 training set sizes. Dashed curves show the scaled expectation value of the distortion magnitude in the AutoAugment policy. [3]

Another approach to learning data augmentation strategies from data is AutoAugment [3], which originally used reinforcement learning to choose a sequence of operations as well as their probability of application and magnitude. More recently, several papers used the AutoAugment search space with improved optimization algorithms to find augmentation policies more efficiently [13, 17, 43, 18]. Although the time it takes to search for policies has been reduced significantly, they still need a separate data augmentation search on a proxy task.

## 3 Systematic failures of a separate proxy task

A central premise of neural architecture search or learned data augmentation is to construct a small proxy task that may be reflective of a larger task [46, 47, 3]. Although this assumption is sufficient for identifying learned augmentation policies to improve performance [3, 45, 26, 17, 13], it is unclear if this assumption is overly stringent and may lead to sub-optimal data augmentation policies. For example, random architectures were found to be competitive with architectures found on small proxy tasks by Yu et al [40].

In this section, we challenge the hypothesis that formulating the problem in terms of a small proxy task is appropriate for learned data augmentation. In particular, we explore this question along two separate dimensions that are commonly restricted to achieve a small proxy task: model size and

dataset size. To study this hypothesis, we systematically measure the effects of data augmentation policies on CIFAR-10 with varying data and model sizes.

In order to systematically compare how the optimal augmentation policy changes under a variety of conditions, we must design an easily alterable augmentation policy. Our augmentation policy used in the below experiments is extremely simple: take $K$ different augmentation operations and for each image during training sample $N$ to apply. Since each augmentation operation has a distortion magnitude (e.g. how many degrees to rotate an image), we utilize the AutoAugment scheme that discretizes each operation magnitude ($M$) from [0, 10]. Now for all $K$ operations we fix them to have the same magnitude $M$. To vary the strength of the augmentation policy we simply change the $M$ or $N$ parameter. More details on the augmentation method can be found in Section 4.

To study how the optimal augmentation policy changes as model size changes, we train a family of Wide-ResNet architectures [41]. The Wide-ResNet model size can be systematically altered through the *widening* parameter governing the number of convolutional filters. For each of these networks, we train the model on CIFAR-10 and measure the final accuracy compared to a baseline model trained with default data augmentations (i.e. horizontal flips and pad-and-crop) [41].

The Wide-ResNet models are all trained with $K$=14 data augmentations over a range of distortion magnitudes $M$ parameterized on a uniform linear scale ranging from [0, 30]. This allows for easily controlling the strength of the data augmentation policies in our experiments.[4]

Figure 1(a) demonstrates the relative gain in accuracy of a model trained across increasing distortion magnitudes ($M$) for three Wide-ResNet models. The squares indicate the distortion magnitude with which achieves the highest accuracy. The results from Figure 1(a) demonstrates a clear systematic trend across distortion magnitudes. In particular, plotting all Wide-ResNet architectures versus the optimal distortion magnitude highlights a clear monotonic trend across increasing network sizes (Figure 1(b)). Namely, larger networks demand larger data distortions for regularization. Figure 2 highlights the visual difference in the optimal distortion magnitude for differently sized models. Conversely, a policy learned on a proxy task (such as AutoAugment) provides a fixed distortion magnitude (Figure 1b, dashed line) for all architectures that is clearly sub-optimal.

The second dimension for constructing a small proxy task is to train the proxy on a small subset of the training data. Figure 1(c) demonstrates the relative gain in accuracy of Wide-ResNet-28-10 trained across increasing distortion magnitudes for varying amounts of CIFAR-10 training data. The squares indicate the distortion magnitude that achieves the highest accuracy. From Figure 1(c) we observe that models trained on smaller training dataset sizes may gain more improvement from data augmentation (e.g. 3.0% versus 1.5% in Figure 1(c)). Furthermore, it appears that the optimal distortion magnitude is larger for models that are trained on larger datasets. At first glance, this may disagree with the expectation that smaller datasets require stronger regularization.

Figure 1(d) demonstrates that the optimal distortion magnitude increases monotonically with training set size. One hypothesis for this counter-intuitive behavior is that aggressive data augmentation leads to a low signal-to-noise ratio in small datasets. Regardless, this trend highlights the need for increasing the strength of data augmentation on larger datasets. This reveals a shortcoming of optimizing learned augmentation policies on a proxy task since it is comprised of a subset of the training data. Namely, the learned augmentation may learn an augmentation strength more tailored to the proxy task instead of the larger task of interest.

The dependence of augmentation strength on the dataset and model size indicate that a small proxy task may provide a sub-optimal indicator of performance on a larger task. This empirical result suggests that a distinct strategy may be necessary for finding an optimal data augmentation policy. Figure 1 suggest that merely searching for a shared distortion magnitude $M$ across all transformations may provide sufficient gains that exceed learned optimization methods using proxy tasks. Additionally, we see that optimizing individual magnitudes instead of tying them into a single parameter leads to minor improvement in performance (see Section A.2 in Appendix).

Furthermore, Figure 1(a) and 1(c) indicate that merely sampling a few distortion magnitudes is sufficient to achieve good results. Coupled with a second free parameter $N$, we consider these results to prescribe an algorithm for learning an augmentation policy. In the subsequent sections, we

identify two free parameters $N$ and $M$ that specify our new augmentation method: RandAugment. We compare RandAugment, that just requires a simple grid search[5], against computationally-heavy learned data augmentations based on proxy tasks.

## 4    Automated data augmentation without a proxy task

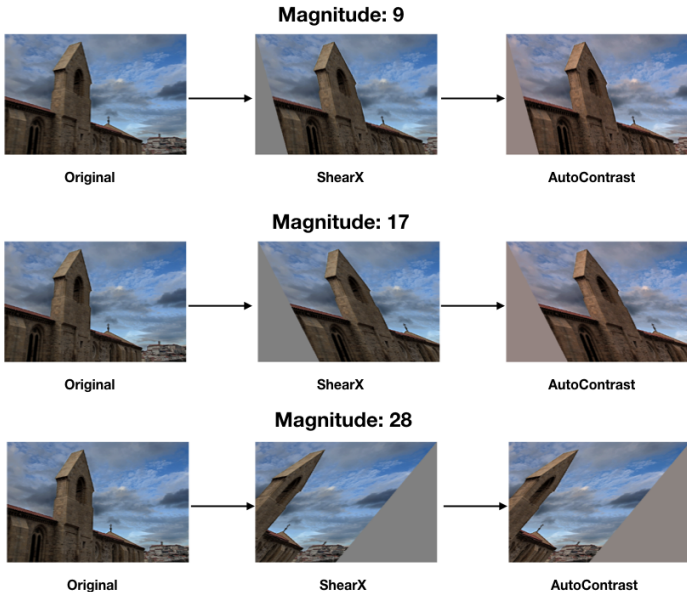

Figure 2: **Example images augmented by RandAugment.** In these examples $N$=2 and three magnitudes are shown corresponding to the optimal distortion magnitudes for ResNet-50, EfficientNet-B5 and EfficientNet-B7, respectively. As the distortion magnitude increases, the strength of the augmentation increases.

The primary goal of RandAugment is to remove the need for a separate search phase on a proxy task. The reason we wish to remove the search phase is because a separate search phase significantly complicates training and is computationally expensive. More importantly, the proxy task may provide sub-optimal results, as seen in the previous section. In order to remove a separate search phase, we aspire to fold the parameters for the data augmentation strategy into the hyper-parameters for training a model. Given that previous learned augmentation methods contained 30+ parameters [3, 17, 13], we focus on vastly reducing the parameter space for data augmentation.

Previous work indicates that the main benefit of learned augmentation policies arise from increasing the diversity of examples [3, 13, 17]. Indeed, previous work enumerated a policy in terms of choosing which transformations to apply out of $K$=14 available transformations, and probabilities for applying each transformation: `identity`, `autoContrast`, `equalize`, `rotate`, `solarize`, `color`, `posterize`, `contrast`, `brightness`, `sharpness`, `shear-x`, and `shear-y`.

In order to reduce the parameter space but still maintain image diversity, we replace the learned policies and probabilities with a simplified procedure. For each image we applying each transformation with a parameter-free procedure of *always* selecting a transformation with uniform probability $\frac{1}{K}$. Given $N$ transformations for a training image, RandAugment may thus express $K^N$ potential outcomes (policies). The final set of parameters to consider is the magnitude of the each augmentation distortion. Following [3], we employ the same linear scale for indicating the strength of each transformation. Briefly, each transformation resides on an integer scale from 0 to 10 where a value of 10 indicates the maximum scale for a given transformation. A data augmentation policy consists of identifying an integer for each augmentation operation [3, 17, 13]. In order to reduce the parameter space further, we observe that the learned magnitude for each transformation follows a similar schedule during training (e.g. Figure 4 in [13]) and postulate that a *single* global distortion

```
transforms = [
"Identity", "AutoContrast", "Equalize", "Rotate", "Solarize",
"Color", "Posterize", "Contrast", "Brightness", "Sharpness",
"ShearX", "ShearY", "TranslateX", "TranslateY"]

def randaugment(N, M):
"""Generate a set of distortions.

  Args:
    N: Number of augmentation transformations to apply
        sequentially.
    M: Magnitude for all the transformations.
"""

  sampled_ops = np.random.choice(transforms, N)
  return [(op, M) for op in sampled_ops]
```

Figure 3: Python code for RandAugment based on NumPy.

$M$ may suffice for parameterizing all transformations. We experimented with four methods for the schedule of $M$ during training: constant magnitude, random magnitude, a linearly increasing magnitude, and a random magnitude with increasing upper bound. The details of this experiment can be found in Appendix A.1.

The resulting algorithm contains two parameters $N$ and $M$ and may be expressed simply in two lines of Python code (Figure 3). Both parameters are human-interpretable such that larger values of $N$ and $M$ increase regularization strength. Standard methods may be employed to efficiently perform hyperparameter optimization [31], however given the extremely small search space we find that naive grid search is quite effective (Section 3). We justify all of the choices of this proposed algorithm in the subsequent sections by comparing the efficacy of our simple method to previous learned data augmentation methods.

## 5   Experiments

To explore the space of data augmentations, we experiment with core image classification and object detection tasks. In particular, we focus on CIFAR-10, CIFAR-100, SVHN, and ImageNet datasets as well as COCO object detection so that we may compare with previous work. For all datasets, we replicate the corresponding architectures and set of data transformations. Our goal is to demonstrate the relative benefits of employing this method over previous learned augmentation methods; the RandAugment model and the baseline model do not differ in any setting other than the data augmentation strategy.

**CIFAR-10** has been extensively studied with previous data augmentation methods, so we first test this proposed method on it. The default augmentations for all methods include flips, pad-and-crop and Cutout [6]. $N$ and $M$ were selected based on the validation performance on 5K held out examples from the training set for 1 and 5 settings for $N$ and $M$, respectively. Results indicate that despite its simplicity, RandAugment achieves competitive results on CIFAR-10 across four network architectures (Table 2). As a more challenging task, we additionally compare the efficacy of RandAugment on CIFAR-100 for Wide-ResNet-28-2 and Wide-ResNet-28-10. On the held out 5K dataset, we sampled 2 and 4 settings for $N$ and $M$, respectively (i.e. $N$={1, 2} and $M$={2, 6, 10, 14}). For Wide-ResNet-28-2 and Wide-ResNet-28-10, we find that $N$=1, $M$=2 and $N$=2, $M$=14 achieves best results, respectively. Again, RandAugment achieves competitive or superior results compared to AutoAugment across both architectures (Table 2).

**SVHN** has a core training set of 73K images [24]. In addition, SVHN contains 531K less difficult "extra" images to augment training. Because SVHN is composed of numbers instead of natural images, the data augmentation strategy for SVHN may differ substantially from CIFAR-10. Indeed, [3] identified a qualitatively different policy for CIFAR-10 than SVHN. Likewise, in a semi-supervised setting for CIFAR-10, a policy learned from CIFAR-10 performs better than a policy learned from SVHN [37]. We compare the performance of the augmentation methods on SVHN with and without the extra data on Wide-ResNet-28-2 and Wide-ResNet-28-10 (Table 2). In spite of the large differences between SVHN and CIFAR, RandAugment consistently matches or outperforms previous

|  | baseline | PBA | Fast AA | OHL AA | Adv AA | AA | RA |
|---|---|---|---|---|---|---|---|
| **CIFAR-10** | | | | | | | |
| Wide-ResNet-28-2 | 94.9 | - | - | - | - | **95.9** | 95.8 |
| Wide-ResNet-28-10 | 96.1 | 97.4 | 97.3 | 97.4 | **98.1** | 97.4 | 97.3 |
| Shake-Shake | 97.1 | 98.0 | 98.0 | - | **98.1** | 98.0 | 98.0 |
| PyramidNet | 97.3 | 98.5 | 98.3 | - | **98.6** | 98.5 | 98.5 |
| **CIFAR-100** | | | | | | | |
| Wide-ResNet-28-2 | 75.4 | - | - | - | - | **78.5** | 78.3 |
| Wide-ResNet-28-10 | 81.2 | 83.3 | 82.7 | - | **84.5** | 82.9 | 83.3 |
| **SVHN (core set)** | | | | | | | |
| Wide-ResNet-28-2 | 96.7 | - | - | - | - | 98.0 | **98.3** |
| Wide-ResNet-28-10 | 96.9 | - | - | - | - | 98.1 | **98.3** |
| **SVHN** | | | | | | | |
| Wide-ResNet-28-2 | 98.2 | - | - | - | - | **98.7** | **98.7** |
| Wide-ResNet-28-10 | 98.5 | 98.9 | 98.8 | - | - | 98.9 | **99.0** |

Table 2: **Test accuracy (%) on CIFAR-10, CIFAR-100, SVHN and SVHN core set**. Comparisons across default data augmentation (baseline), Population Based Augmentation (PBA) [13], Fast AutoAugment (Fast AA) [17], Online Hyper-parameter Learning for Auto-Augmentation Strategy (OHL AA) [18], Adversarial AutoAugment (Adv AA) [43], AutoAugment (AA) [3] and proposed RandAugment (RA). Note that baseline and AA are replicated in this work. SVHN core set consists of 73K examples. The Shake-Shake model [8] employed a 26 2×96d configuration, and the PyramidNet model used the ShakeDrop regularization [38]. Results reported by us are averaged over 10 independent runs. Bold indicates best results.

methods with no alteration to the list of transformations employed. Notably, for Wide-ResNet-28-2, applying RandAugment to the core training dataset improves performance more than augmenting with 531K additional training images (98.3% vs. 98.2%). For Wide-ResNet-28-10, RandAugment is competitive with augmenting the core training set with 531K training images (i.e. within 0.2%). Nonetheless, Wide-ResNet-28-10 with RandAugment matches the previous state-of-the-art accuracy on SVHN which used a more advanced model [3].

**ImageNet:** Data augmentation methods that improve CIFAR-10 and SVHN models do not always improve large-scale tasks such as ImageNet. For instance, Cutout substantially improves CIFAR and SVHN performance [6], but fails to improve ImageNet [23]. Likewise, AutoAugment does not increase the performance on ImageNet as much as other tasks [3], especially for large networks (e.g. +0.4% for AmoebaNet-C [3]). One plausible reason for the lack of strong gains is that the small proxy task was particularly impoverished by restricting the task to ∼10% of the 1000 ImageNet classes.

|  | baseline | Fast AA | AA | RA |
|---|---|---|---|---|
| ResNet-50 | 76.3 | **77.6** | **77.6** | **77.6** |
| EfficientNet-B5 | 83.1 | - | 83.6 | **83.7** |
| EfficientNet-B7 | 83.7 | - | 84.3 | **84.7** |

Table 3: **ImageNet results.** Top-1 accuracy (%) on ImageNet. Baseline and AutoAugment (AA) results on ResNet-50 are from [3]. Fast AutoAugment (Fast AA) results are from [17]. Note that the ResNet-50 results for all augmentation methods used the baseline model with the same performance, which we reproduced in this paper. EfficientNet results with and without AutoAugment are from [34]. Highest accuracy for each model is presented in bold.

Table 3 compares the performance of RandAugment to other learned augmentation approaches on ImageNet. RandAugment matches the performance of AutoAugment and Fast AutoAugment on the smallest model (ResNet-50), but on larger models RandAugment outperforms other methods achieving increases of up to +1.0% above the baseline. For instance, on EfficientNet-B7, the resulting model achieves 84.7% exhibiting a 1.0% improvement over the baseline augmentation. These systematic gains are similar to the improvements achieved with engineering new architectures [47, 21], however these gains arise without incurring additional computational cost at inference time.

**Object detection with COCO dataset:** To further test the generality of this approach, we next explore a related task of large-scale object detection on the COCO dataset [20]. Learned augmentation policies have improved object detection and lead to state-of-the-art results [45]. We followed

| augmentation | search space | ResNet-101 | ResNet-200 |
|---|---|---|---|
| Baseline | 0 | 38.8 | 39.9 |
| AutoAugment | $10^{34}$ | **40.4** | **42.1** |
| RandAugment | $10^{2}$ | 40.1 | 41.9 |

Table 4: **Results on object detection.** Mean average precision (mAP) on COCO detection task. Search space size is reported as the order of magnitude of the number of possible augmentation policies. Models are trained for 300 epochs from random initialization following [45].

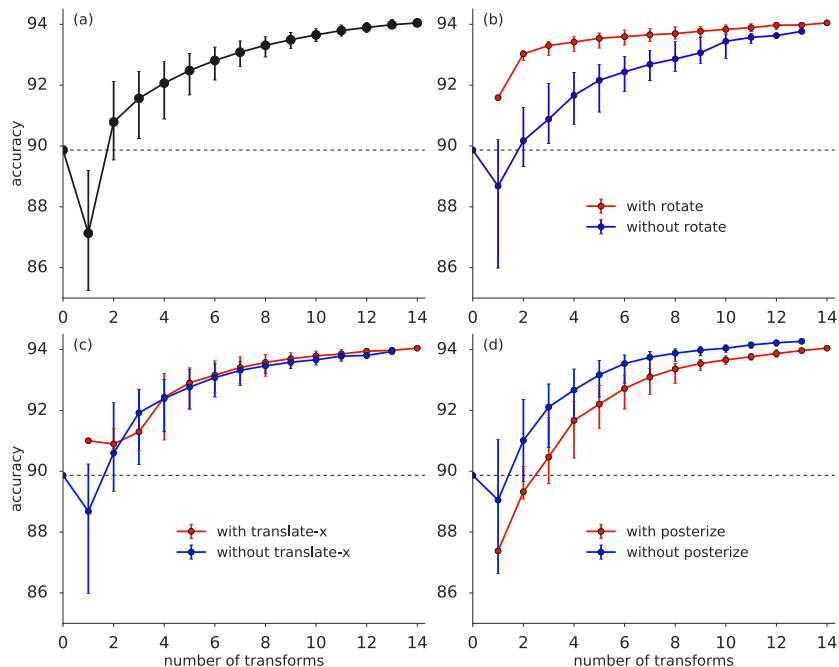

Figure 4: **Average performance improves when more transformations are included in RandAugment.** All panels report median CIFAR-10 validation accuracy for Wide-ResNet-28-2 model architectures [41] trained with RandAugment ($N = 3$, $M = 4$) using randomly sampled subsets of transformations. No other data augmentation is included in training. Error bars indicate 30th and 70th percentile. (a) Median accuracy for randomly sampled subsets of transformations. (b) Median accuracy for subsets with and without the `Rotate` transformation. (c) Median accuracy for subsets with and without the `translate-x` transformation. (d) Median accuracy for subsets with and without the `posterize` transformation. Dashed curves show the accuracy of the model trained without any augmentations.

previous work by training on the same architectures and following the same training schedules (see Appendix A.4.4). Briefly, we employed RetinaNet [19] with ResNet-101 and ResNet-200 as a backbone [10]. Models were trained for 300 epochs from random initialization.

Table 4 compares results between a baseline model, AutoAugment and RandAugment. AutoAugment leveraged additional, specialized transformations not afforded to RandAugment in order to augment the localized bounding box of an image [45]. In addition, note that AutoAugment expended ∼15K GPU hours for search, where as RandAugment was tuned over only 6 values of the hyper-parameters (see Appendix A.4.4). In spite of the smaller library of specialized transformations and the lack of a separate search phase, RandAugment surpasses the baseline model and provides competitive accuracy with AutoAugment. We reserve for future work to expand the transformation library to include bounding box specific transformation to potentially improve RandAugment results even further.

**Investigating the dependence on the included transformations:** RandAugment achieves state-of-the-art results across different tasks and datasets using the same list of transformations. This result suggests that RandAugment is largely insensitive to the selection of transformations for different datasets. To further study the sensitivity, we experimented with RandAugment on a Wide-ResNet-

28-2 trained on CIFAR-10 for randomly sampled subsets of the full list of 14 transformations. We did not use flips, pad-and-crop, or cutout to only focus on the improvements due to RandAugment with random subsets of augmentation operations. Figure 4 (a) suggests that the median validation accuracy due to RandAugment improves as the number of transformations is increased. However, even with only two transformations, RandAugment leads to more than 1% improvement in validation accuracy on average.

To get a sense for the effect of individual transformations, we calculate the average improvement in validation accuracy for each transformation when they are added to a random subset of transformations. We list the transformations in order of most helpful to least helpful in Table 5. We see that while geometric transformations individually make the most difference, some of the color transformations lead to a degradation of validation accuracy on average. The transformation `rotate` is most helpful on average, which was also observed previously [3, 45]. To see the effect of representative transformations in more detail, we repeat the analysis in Figure 4a for subsets with and without (`rotate`, `translate-x`, and `posterize`). Surprisingly, `rotate` can significantly improve performance and lower variation even when included in small subsets of RandAugment transformations, while `posterize` seems to hurt all subsets of all sizes.

| transformation | $\Delta$ (%) | transformation | $\Delta$ (%) |
|---|---|---|---|
| rotate | **1.3** | shear-x | 0.9 |
| shear-y | 0.9 | translate-y | 0.4 |
| translate-x | 0.4 | autoContrast | 0.1 |
| sharpness | 0.1 | identity | 0.1 |
| contrast | 0.0 | color | 0.0 |
| brightness | 0.0 | equalize | -0.0 |
| solarize | -0.1 | posterize | -0.3 |

Table 5: **Average improvement due to each transformation.** Average difference in validation accuracy (%) when a particular transformation is added to a randomly sampled set of transformations. For this ablation study, Wide-ResNet-28-2 models were trained on CIFAR-10 using RandAugment ($N = 3$, $M = 4$) with the randomly sampled set of transformations, with no other data augmentation. Note that while Table 5 shows the average effect of adding individual transformations to randomly sampled subsets of transformations, Figure 4a shows that including all transformations together leads to a good result.

## 6   Discussion

Data augmentation is a necessary method for achieving state-of-the-art performance [30, 15, 5, 42, 9, 26]. Learned data augmentation strategies have helped automate the design of such strategies and likewise achieved state-of-the-art results [3, 17, 13, 45]. In this work, we demonstrated that previous methods of learned augmentation suffers from systematic drawbacks. Namely, not tailoring the number of distortions and the distortion magnitude to the dataset size nor the model size leads to sub-optimal performance. To remedy this situation, we propose a simple parameterization for targeting augmentation to particular model and dataset sizes. We demonstrate that RandAugment is competitive with or outperforms previous approaches [3, 17, 13, 45] on CIFAR-10/100, SVHN, ImageNet and COCO without a separate search for data augmentation policies.

In previous work, scaling learned data augmentation to larger dataset and models have been a notable obstacle. For example, AutoAugment and Fast AutoAugment could only be optimized for small models on reduced subsets of data [3, 17]; PBA was not reported for large-scale problems [13]. The proposed method RandAugment scales quite well to datasets such as ImageNet and COCO while incurring minimal computational cost (e.g. 2 hyper-parameters), but notable predictive performance gains. Future work will study how this method applies to other machine learning domains, where data augmentation is known to improve predictive performance, such as image segmentation [1], 3-D perception [25], speech recognition [12] or audio recognition [11]. In particular, we wish to better understand if or when datasets or tasks may require a separate search phase to achieve optimal performance. Finally, an open question remains how one may tailor the set of transformations for a particular task.

## Broader Impact

Data augmentation has recently played a central role in a variety of deep learning research directions, including self- and semi-supervised learning, out-of-domain generalization and calibration, and reinforcement learning. Being able to improve data augmentation pipelines efficiently has promise for more accurate, robust, and calibrated machine learning models to be used for positive societal applications (e.g. autonomous vehicles, healthcare, etc.). On the other hand, this improved technology can also be used by bad actors. Finally, removing the need for a separate augmentation search phase reduces the carbon footprint and computational cost of training better models.

## Acknowledgments and Disclosure of Funding

We thank Samy Bengio, Daniel Ho, Ildoo Kim, Jaehoon Lee, Zhaoqi Leng, Hanxiao Liu, Raphael Gontijo Lopes, Ruoming Pang, Ben Poole, Mingxing Tan, and the rest of the Brain team for their help. Google is the sole source of funding for this work.

## Footnotes

[2] github.com/tensorflow/tpu/tree/master/models/official/efficientnet

[3] github.com/tensorflow/tpu/tree/master/models/official/resnet

[4]Note that the range of magnitudes exceeds the specified range of magnitudes in the Methods because we wish to explore a larger range of magnitudes for this preliminary experiment. We retain the same scale as [3] for a value of 10 to maintain comparable results.

[5]We found that random search instead of grid search performs as well for optimizing the distortion magnitude.

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
