[Supplementary Material]

# A  Appendix

## A.1  Magnitude methods

| Magnitude Method | Accuracy |
|---|---|
| Random Magnitude | 97.3 |
| Constant Magnitude | 97.2 |
| Linearly Increasing Magnitude | 97.2 |
| Random Magnitude with Increasing Upper Bound | 97.3 |

Table 6: **Results for different ways of setting the global magnitude parameter $M$.** All magnitude methods were run on CIFAR-10 with Wide-ResNet-28-10 for 200 epochs. The reported accuracy is the average of 10 runs on the validation set for the best hyperparamter setting for that magnitude method. All magnitude methods searched over had 48 different hyperparameter settings tried.

Here we study four different methods for parameterizing the distortion parameter $M$. Four different methods were explores: Random Magnitude, Constant Magnitude, Linearly Increasing Magnitude and Random Magnitude with Increasing Upper Bound. The Random Magnitude method uniformly randomly samples the distortion magnitude between two values. Constant Magnitude sets the distortion magnitude to a constant number during the course of training. Linearly Increasing Magnitude interpolates the distortion magnitude during training between two values (going from low to high). Random Magnitude with Increasing Upper Bound is similar to Random Magnitude, but the upper bound is increased linearly during training. In preliminary experiments, we found that all strategies worked equally well. Thus, we selected a constant magnitude because this strategy includes only a single hyper-parameter, and we employ this for the rest of the work. The results from our experiment on trying the different magnitude strategies can be see in Table 6.

## A.2  Optimizing individual transformation magnitudes

Figure 5: **Performance when magnitude is changed for one image transformation.** This plot uses a shared magnitude for all image transformations and then changes the magnitude of only one operation while keeping the others fixed. Two different architectures were tried (WRN-28-2 and WRN-28-10) and two different image transformations were changed (Rotate and TranslateX), which results in the 4 lines shown. Twenty different magnitudes were tried for the selected transformation ($[0-19]$). The squares indicate the optimal magnitude found and the diamonds indicate the magnitude used for all other transformations (4 for WRN-28-2 and 5 for WRN-28-10).

Figure 5 demonstrates that changing the magnitude for one transformation, when keeping the rest fixed results in a very minor accuracy change. This suggests that tying all magnitudes together into a single value $M$ is not greatly hurting the model performance. Across all for settings in Figure 5 the difference in accuracy of the tied magnitude vs the optimal one found was 0.19% 0.18% for the rotation operation experiments and 0.07% 0.05% for the TranslateX experiments. Changing one transformation does not have a huge impact on performance, which leads us to think that tying all magnitude parameters together is a sensible approach that drastically reduces the size of the search-space.

## A.3 Learning the probabilities for selecting image transformations using gradients

|  | baseline | AA | RA | + 1st |
|---|---|---|---|---|
| **Reduced CIFAR-10** | | | | |
| Wide-ResNet-28-2 | 82.0 | **85.6** | 85.3 | 85.5 |
| Wide-ResNet-28-10 | 83.5 | **87.7** | 86.8 | 87.4 |
| **CIFAR-10** | | | | |
| Wide-ResNet-28-2 | 94.9 | 95.9 | 95.8 | **96.1** |
| Wide-ResNet-28-10 | 96.1 | **97.4** | 97.3 | **97.4** |

Table 7: **Differentiable optimization for augmentation can improve RandAugment.** Test accuracy (%) from differentiable RandAugment for reduced (4K examples) and full CIFAR-10. The $1^{st}$-order approximation ($1^{st}$) is based on density matching (Section A.3). Models trained on reduced CIFAR-10 were trained for 500 epochs. CIFAR-10 models trained using the same hyperparameters as previous. Each result is averaged over 10 independent runs.

RandAugment selects all image transformations with equal probability. This opens up the question of whether learning $K$ probabilities may improve performance further. Most of the image transformations (except posterize, equalize, and autoContrast) are differentiable, which permits back-propagation to learn the $K$ probabilities [22]. Let us denote $\alpha_{ij}$ as the learned probability of selecting image transformation $i$ for operation $j$. For $K$=14 image transformations and $N$=2 operations, $\alpha_{ij}$ constitutes 28 parameters. We initialize all weights such that each transformation is equal probability (i.e. RandAugment), and update these parameters based on how well a model classifies a held out set of validation images distorted by $\alpha_{ij}$. This approach was inspired by density matching [17], but instead uses a differentiable approach in lieu of Bayesian optimization. We label this method as a $1^{st}$-order density matching approximation.

To test the efficacy of density matching to learn the probabilities of each transformation, we trained Wide-ResNet-28-2 and Wide-ResNet-28-10 on CIFAR-10 and the reduced form of CIFAR-10 containing 4K training samples. Table 7 indicates that learning the probabilities $\alpha_{ij}$ slightly improves performance on reduced and full CIFAR-10 (RA vs $1^{st}$). The $1^{st}$-order method improves accuracy by more than 3.0% for both models on reduced CIFAR-10 compared to the baseline of flips and pad-and-crop. On CIFAR-10, the $1^{st}$-order method improves accuracy by 0.9% on the smaller model and 1.2% on the larger model compared to the baseline. We further see that the $1^{st}$-order method always performs better than RandAugment, with the largest improvement on Wide-ResNet-28-10 trained on reduced CIFAR-10 (87.4% vs. 86.8%). On CIFAR-10, the $1^{st}$-order method outperforms AutoAugment on Wide-ResNet-28-2 (96.1% vs. 95.9%) and matches AutoAugment on Wide-ResNet-28-10. Although the density matching approach is promising, this method can be expensive as one must apply all $K$ transformations $N$ times to each image independently. Hence, because the computational demand of $KN$ transformations is prohibitive for large images, we reserve this for future exploration. In summary, we take these results to indicate that learning the probabilities through density matching may improve the performance on small-scale tasks and reserve explorations to larger-scale tasks for the future.

## A.4 Experimental Details

### A.4.1 CIFAR

The Wide-ResNet models were trained for 200 epochs with a learning rate of 0.1, batch size of 128, weight decay of 5e-4, and cosine learning rate decay. Shake-Shake [8] model was trained for 1800 epochs with a learning rate of 0.01, batch size of 128, weight decay of 1e-3, and cosine learning rate

decay. ShakeDrop [38] models were trained for 1800 epochs with a learning rate of 0.05, batch size of 64 (as 128 did not fit on a single GPU), weight decay of 5e-5, and cosine learning rate decay.

On CIFAR-10, we used 3 for the number of operations applied ($N$) and tried 4, 5, 7, 9, and 11 for magnitude. For Wide-ResNet-2 and Wide-ResNet-10, we find that the optimal magnitude is 4 and 5, respectively. For Shake-Shake (26 2x96d) and PyramidNet + ShakeDrop models, the optimal magnitude was 9 and 7, respectively.

### A.4.2 SVHN

For both SVHN datasets, we applied cutout after RandAugment as was done for AutoAugment and related methods. On core SVHN, for both Wide-ResNet-28-2 and Wide-ResNet-28-10, we used a learning rate of 5e-3, weight decay of 5e-3, and cosine learning rate decay for 200 epochs. We set $N = 3$ and tried 5, 7, 9, and 11 for magnitude. For both Wide-ResNet-28-2 and Wide-ResNet-28-10, we find the optimal magnitude to be 9.

On full SVHN, for both Wide-ResNet-28-2 and Wide-ResNet-28-10, we used a learning rate of 5e-3, weight decay of 1e-3, and cosine learning rate decay for 160 epochs. We set $N = 3$ and tried 5, 7, 9, and 11 for magnitude. For Wide-ResNet-28-2, we find the optimal magnitude to be 5; whereas for Wide-ResNet-28-10, we find the optimal magnitude to be 7.

### A.4.3 ImageNet

The ResNet models were trained for 180 epochs using the standard ResNet-50 training hyperparameters. The image size was 224 by 224, the weight decay was 0.0001 and the momentum optimizer with a momentum parameter of 0.9 was used. The learning rate was 0.1, which gets scaled by the batch size divided by 256. A global batch size of 4096 was used, split across 32 workers. For ResNet-50 the optimal distortion magnitude was 9 and ($N = 2$). The distortion magnitudes we tried were 5, 7, 9, 11, 13, 15 and the values of $N$ that were tried were 1, 2 and 3.

The EfficientNet experiments used the default hyper parameters and training schedule, which can be found in [34]. We trained for 350 epochs, used a batch size of 4096 split across 256 replicas. The learning rate was 0.016, which gets scaled by the batch size divided by 256. We used the RMSProp optimizer with a momentum rate of 0.9, epsilon of 0.001 and a decay of 0.9. The weight decay used was 1e-5. For EfficientNet B5 the image size was 456 by 456 and for EfficientNet B7 it was 600 by 600. For EfficientNet B5 we tried $N = 2$ and $N = 3$ and found them to perform about the same. We found the optimal distortion magnitude for B5 to be 17. The different magnitudes we tried were 8, 11, 14, 17, 21. For EfficientNet B7 we used $N = 2$ and found the optimal distortion magnitude to be 28. The magnitudes tried were 17, 25, 28, 31.

The default augmentation of horizontal flipping and random crops were used on ImageNet, applied before RandAugment. The standard training and validation splits were employed for training and evaluation.

### A.4.4 COCO

We applied horizontal flipping and scale jitters in addition to RandAugment. We used the same list of data augmentation transformations as we did in all other classification tasks. Geometric operations transformed the bounding boxes the way it was defined in Ref. [45]. We used a learning rate of 0.08 and a weight decay of $1e-4$. The focal loss parameters are set to be $\alpha = 0.25$ and $\gamma = 1.5$. We set $N = 1$ and tried distortion magnitudes between 4 and 9. We found the optimal distortion magnitude for ResNet-101 and ResNet-200 to be 5 and 6, respectively.