[Reviews · NeurIPS 2020]

Review 1

Summary and Contributions: This paper tackles automated data augmentation by first identifying the problem of policy search on proxy task, a standard practice in the field. It then propose a reduced search space to find good data augmentation strategies. The resultant search space is simple enough for grid search yet still yield competitive performance compared to other automated data augmentation methods.

Strengths: The paper provides a very insightful analysis on how the data augmentation policy search is sub-optimal if they are performed on proxy tasks. More precisely, as the authors point out quite accurately, the proxy tasks are typically smaller dataset/smaller model. As shown in Figure 1, these two dimensions both have a non-trivial (and sometimes counter-intuitive) impact on the final accuracy, showing search on proxy tasks may not be a good strategy. I find the information (both this analysis, and the proposed simple search space) provided by this paper useful to demystify important aspects of auto data augmentation, and perhaps can provide inspiration for deeper insights of other applications of AutoML as well. More precisely, most of the AutoML based methods (auto data augmentation in particular) boast about their “flexibility” while making drastic simplifications. But if the flexibility comes with a large search space that require approximation, could the approximation be so bad that the large search space is simply useless? This paper certainly provides a concrete thesis that caution against this trend. In practice, this paper propose a much simpler algorithm compared to existing auto data augmentation algorithm that can achieve SOTA results on classification/detection tasks. This by itself is a good contribution.

Weaknesses: The paper only shows proxy task + complicated search space may not work as well as using a simple search task without much approximation. It doesn’t really tell us what happens if a complicated search space can be efficiently explored on the real task. In this sense, this paper is only a reflection of current practice, without providing a clear direction forward. In fact, the simplification of this paper (reducing the search space to number of op to apply, and the shared magnitude of ops) seems like an over-kill. By doing that, it misses an opportunity to answer some interesting question, such as: “Does assigning a different magnitude to different ops useful at all in auto data augmentation”? Intuitively, there should be an intermediate search space that is perhaps simpler than most used in the literature, but still simple enough so that the search can be done on the target task. In a sense, this paper seems to mix two related, but distinct goals prematurely. Goal 1: have an efficient search algorithm on target task (not proxy task). Goal 2: Replace an approximate search algorithm (such as RL used in AutoAugment) with a grid search. While goal 1 is clearly justified by careful analysis, goal 2 seems like an after-thought. Update: The rebuttal addresses most of my most of my concerns. I think it is a on balance a good paper and should be accepted. I have revised my score to "accept". I have read the other reviews and compare notes with the rebuttal. I agree with the authors in that the paper shouldn't be discounted as not-novel. My reasoning is that if data augmentation (with the current design in augmentation strategies) does not really need a full AutoML solution, then a paper that finds this empirically has its values.

Correctness: The paper has good empirical analysis. The comparisons seems correct and can justify the conclusions.

Clarity: The paper is in general well-written. However, there are a few problems that should be addressed Line 158-160: Why can RandAugment represent K^N policies? If it only searches N, wouldn’t it mean it can only search among N options? Perhaps it should be K^N outcomes? Line 167-168: What does M mean precisely? How does the search in M work, is it just selecting which schedule to use? What is the range of M and N, the paper doesn’t seem to make this clear.

Relation to Prior Work: The literature survey is comprehensive and useful.

Reproducibility: No

Additional Feedback:


Review 2

Summary and Contributions: This paper proposes an efficient method to search for data augmentation policies. It conducts a simple grid search on a vastly reduced search space. The experimental results show the competitive performances with some previous work on multiple tasks.

Strengths: 1. This work empirically studies the relationship between the optimal strength of data augmentation and the model size/training set size. The observation indicates that proxy-task-based search methods may be sub-optimal for learning and transferring augmentation policies. 2. This work designs a small policy search space. It significantly reduces the computation cost, getting rid of a separate expensive search phase and proxy tasks. 3. Experimental results are shown for CIFAR-10/100, SVHN, ImageNet, and COCO datasets. This method achieves equal or better performance over some previous methods.

Weaknesses: The mothod of this paper are not so complecate. The author made the point clear that their work decovers the power of the random augmentation. However, some how I think some conventional augmentation procedure doesn't seems entirely another category. Personaly, I would like to see more inovention in this work. 2. This work uses a much smaller search space than much previous work, resulting in fixed searched policies. However, PBA [2] points out that an augmentation function that can reduce generalization error at the end of training is not necessarily a good function at initial phases. So the end goal of PBA is to learn a schedule of augmentation policies as opposed to a fixed policy. Besides, previous works [3,4] also replaces the fixed augmentation policy with a dynamic schedule of augmentation policy along with the training process, and achieves better performance than this paper with competitive computational costs. All of these may indicate the limitations of the fixed policies. Reference: [1] Cubuk, Ekin D., et al. "Autoaugment: Learning augmentation strategies from data." Proceedings of the IEEE conference on computer vision and pattern recognition. 2019. [2] Ho, Daniel, et al. "Population based augmentation: Efficient learning of augmentation policy schedules." International Conference on Machine Learning. 2019. [3] Zhang, Xinyu, et al. "Adversarial autoaugment."International Conference on Learning Representations . 2020. [4] Lin, Chen, et al."Online hyper-parameter learning for auto-augmentation strategy" Proceedings of the IEEE International Conference on Computer Vision. 2019.

Correctness: A claim in L013 may be controversial ("our method achieves equal or better performance over previous automated augmentation strategies"). Because another previous works, achieves competitive performance with this paper.

Clarity: The paper is well written and easy to follow.

Relation to Prior Work: Yes, it clearly discussed how this work differs from some previous automated data augmentation work. However, several strongly related works are missing in their reference (see weekness reference [3,4]).

Reproducibility: Yes

Additional Feedback: Update: The rebuttal addresses some of my concerns. I have revised my score. I think overall this work gives a pratical implemation of better augmentation compared with current deafult settings and could increase performance of baseline models.


Review 3

Summary and Contributions: The author proposed a novel automated data augmentation method named "RandAugment". It designed a vastly smaller search space than the previous works, thus reducing the computation expense. Moreover, in the proposed search space, the author used a random search strategy that achieved some promising results on classification and object detection tasks in automated data augmentation. I have read the rebuttal and would keep the score.

Strengths: 1. Comparing to previous works, the proposed method could reduce the search space from 10^32 to only 100. Moreover, the proposed work is easy to implement thanks to the random search or grid search. 2. The ablation experiments seems ok and adequately designed. It's interesting that posterize has a negative impact on these datasets and rotate\shear\translate are the most effective transformation which is consistent with common sense. 3. The experiments verify that the proxy task may provide sub-optimal results and the proposed method could be directly used on large datasets. Furthermore, RandAugment is largely insensitive to the selection of transformations for different datasets. 4. Paper is well written and easy to understand.

Weaknesses: 1. Although experimental results show the effect of the proposed method, the contribution of this work may be incremental. The idea of this paper seems to come from the paper "EVALUATING THE SEARCH PHASE OF NEURAL ARCHITECTURE SEARCH" that random search performs better than elaborate search strategies. Could you highlight your main contributions and the differerence with it, as well as why such differences make sense? 2. It's encouraged to give the details of the search space of previous works like AA, Fast AA, and PBA. 3. It would be better to place the time cost in the other tables of experiments like in Table 1.

Correctness: Yes

Clarity: Yes

Relation to Prior Work: Yes, the authors show the distinctions between their work and a similar one proposed by E. D. Cubuk et.al.

Reproducibility: Yes

Additional Feedback: 1. There is a typo on page 3, i.e., `discritizes'. 2. The reference [35] lose the paper name.


Review 4

Summary and Contributions: This work proposes a simple and small space to reduce the cost of augmentation search. It samples N transformation operations from the pool, and set a scale M of distortion parameters for all sampled operations. This method leads to significant improvement on multiple tasks like image classification and object detection at minor cost.

Strengths: The paper proposes an extremely simple search space, yet the empirical improvements are significant. It conducts comprehensive empirical experiments with well described training details.

Weaknesses: - Although the empirical study shows good results, the novalty of this paper is limited. - The paper claims a simple grid search is sufficient to get a good result. However a random search might give better result at the same cost, because the optimal parameter may not be included in the grid. - The advantage of this method is obvious when the pool of all transforms is large. Still it is unclear to see whether it is still outstanding when the number of transformation is small. Not all scenarios have a list of 14 augmentations, thus such ablation study is important.

Correctness: Yes

Clarity: The paper is easy to follow.

Relation to Prior Work: Yes

Reproducibility: Yes

Additional Feedback: - Conduct experiment with random search instead of the grid search, and compare the results. Faster convergence and more robust performance is expected from random search given a large pool of augmentations. - Conduct experiment with a smaller pool of transformation candidates and compare with the baselines. It is possible that other augmentation searching algorithms may be comparable or even superior since they might have better results in a reasonable amout of time given a small pool to search from. Update after Rebuttal: I've read the feedback and other reviews. The feedback on novelty is not persuasive enough to me. The statement "...many new semi-supervised and self-supervised papers utilize this method to achieve SOTA..." is more on the side of useful-ness but not the novelty. On the other hand, it is still clear to see that the method brings improvement. Therefore, I'd like to keep my score.

[Author Response · NeurIPS 2020]

We thank the reviewers for the thoughtful comments. We appreciate all of the reviewers found the experiments compelling and the paper was well written.

**R2, R3, R4:** Concerns about novelty. The novelty of our contribution lies in the simplicity of the method. Our proposed method significantly simplifies automated data augmentation search. We emphasize that advances in data augmentation strategies can be as efficacious as advances in architectural changes. Many crucial deep learning improvements have come from simplifying once complex ideas. Furthermore, we show, for the first time, that data augmentation strength depends on training set size with an unintuitive relationship. Finally, while other proposed augmentation methods only work on specific datasets or tasks, our method works on both classification *and* detection. The novelty of our method can also be seen by the fact that many new semi-supervised and self-supervised papers utilize this method to achieve SOTA (e.g. UDA, Noisy Student, FixMatch, ReMixMatch, Tian & Sun et al, Tian & Krishnan et al, Khosla et al.).

**R1:** "different magnitude to different ops?" Thanks for this great suggestion. In the paper we have evaluated if results can be improved by optimizing the magnitudes for different ops individually. Please see Fig.4 in the Appendix, we found that for the larger model it is possible to improve the results by tuning individual magnitudes, but not for the smaller model. "this paper seems to mix two related, but distinct goals" We agree with R1 that it would be interesting to do a more careful analysis on the search phase. We have focused our attention to the search space, and found that when the search space is chosen carefully, the search method can be as simple as grid search. We wanted to make data augmentation as simple as possible for classification and detection, and did not see the need to employ a more complicated search algorithm. "Line 158-160: $K^N$ policies?" R1 is completely right, $K^N$ outcomes would be more correct. We used the term policies here, to place our method in the context of AutoAugment, which would call each of our $K^N$ outcomes a policy. We will clarify this in the text. "Line 167-168: What does M mean precisely" M is the global distortion magnitude that is used by all ops. As R1 mentions, one needs to determine the schedule for M during training. However, as mentioned in Appendix A.1, constant magnitude works as well as other schedules. In order to keep things simple, we used constant magnitude for all experiments in the paper. Then the only decision that needs to be made about $M$ is its constant magnitude, which is optimized for each model as described in text. "range of $M$ and $N$" We listed the values we tried in Section 5 and Appendix A.5. Briefly, we tried $N = \{1, 2, 3\}$ and $M$ between 4 and 28.

**R2:** "it is not surprising that this kind of random augmentation policy could improve the models" We respectfully disagree on this comment. First, our approach is not same as a random policy. AutoAugment [3] evaluated the random policy performance, which is not very good. It would only be worse for larger ImageNet models compared to RandAug (see line 215 for explanation). In our paper, we are not just evaluating a random policy. We are proposing a new search space, which allows even grid search to get SOTA, and discover a positive correlation between training set size and augmentation magnitude. If there are published references that describe such findings, we would love to know. Papers have been published in ICLR/ICML/NEURIPS recently improving the AutoAugment search algorithm. If it were obvious that a simple approach could achieve as good results, we assume those papers would not have been accepted to such leading conferences. PBA suggests dynamic schedules are better We achieve comparable results to PBA ($\pm 0.1\%$) on small datasets (Table 2) while only using fixed policies. On more realistic datasets such as ImageNet and COCO, PBA was *never* evaluated by the authors. In contrast, our method achieves SOTA on ImageNet and strong improvements on COCO. Our method has the added benefit of not requiring a complicated search. We did evaluate dynamic policies (Appendix, Table 5) and found that constant magnitude performs just as well as dynamic schedules and PBA. Thus we see no limitation for fixed policies. Adv. AA and "Online hyperparameter..." comparisons are missing. Adv. AA achieves a better result on CIFAR-10 by 0.1%, however at a significantly increased complexity (which might be the reason they could not evaluate on larger ImageNet models or object detection), and our best result is better on ImageNet by 4.1% with ENet-B8 (they did not evaluate on SOTA architectures, and their policies are not publicly available for comparison). RandAug's strength comes from its ability to scale to large models easily. We do note that these two suggested papers are very impressive, thanks for bringing them up! We will update the paper and L13.

**R3:** "the difference with Yu et al" The cited work focuses on a shared goal of simplifying AutoML, and demonstrates that a random policy may work well for architecture search. In our paper, we do not propose employing a random policy for data augmentation. In fact, random policies were evaluated in the original AutoAugment paper, where it was found that random policies do not perform as well as reinforcement learning. Instead, we propose a new, simplified search space ($10^{32} \rightarrow 10^2$) which outperforms AutoAugment. Note that our analysis also explains why a random policy would not do very well, since it cannot adjust its strength on the model and dataset size. We will cite this paper and discuss its relevance. "details of the search space" We will add more details including the time cost and fix the typos.

**R4:** "Random search vs. grid search" Since our optimization is in 2D, we do not expect there to be a large difference between random search and grid search for magnitude. However we agree that this is a great suggestion, and we will easily add this to the paper. "Ablation for number of transformations" We already ran this ablation (please see Fig.3 in main text and Table 6 in the Appendix). However, we agree that it would be interesting to compare some of these results with the baselines, which we will add to the paper. Thanks again for the great suggestion!

[Meta-Review · NeurIPS 2020]

This paper got mixed reviews. The original ratings are 6,5,5,6. On the positive side, reviewers think the paper solves an important problem. Data augmentation is recognized to be an important step for improving machine learning model performance. However, existing auto data augmentation methods are typically very costly. This paper solves a timely question and has good practical value. On the negative side, reviewers feel the novelty of this paper is somehow limited. Authors did a good job in the response, two reviewers increase their ratings. One reviewer still has concerns on the novelty. AC reads the paper and agree with the reviewers on the positive side. The method proposed in this paper -- reducing the augmentation space -- is useful and this paper will benefit the community. Thus, AC recommends acceptance.